# What Works for Whom? The Influence of Problem Severity, Maladaptive Perfectionism, and Perceived Parental Pressure on the Effectiveness of a School-Based Performance Anxiety Program

**DOI:** 10.3390/bs15040436

**Published:** 2025-03-28

**Authors:** Amanda W. G. van Loon, Hanneke E. Creemers, Simone Vogelaar, Jessica J. Asscher

**Affiliations:** 1Child and Adolescent Studies, Utrecht University, Heidelberglaan 1, 3584 CS Utrecht, The Netherlands; 2Forensic Child and Youth Care Sciences, University of Amsterdam, Nieuwe Achtergracht 127, 1018 WS Amsterdam, The Netherlands; 3Education and Child Studies, Leiden University, Wassenaarseweg 52, 2333 AK Leiden, The Netherlands

**Keywords:** adolescence, school-based intervention programs, moderators, performance anxiety, perfectionism, perceived parental pressure

## Abstract

School-based intervention programs aiming to support adolescents with mental health problems, such as (school-related) stress and performance anxiety, show inconsistent results. In order to make intervention efforts more beneficial, it is crucial to investigate who is most (un)likely to benefit and under what circumstances. The current study aimed to identify whether problem severity, maladaptive perfectionism, and perceived parental pressure moderate the effectiveness of a school-based performance anxiety program, and if this depends on the level of program attendance. The final sample consisted of *N* = 196 adolescents (*M*_age_ = 14.12, *SD* = 0.79, with 53% females) who participated in a randomized controlled trial. ANCOVAs were conducted for two indicators of performance anxiety: test anxiety and fear of failure. The results demonstrated that for test anxiety, the program was only effective for adolescents with higher pretest levels. Moreover, in the subsample of adolescents with higher program attendance, the program was only effective for adolescents with higher self-criticism perfectionism, and larger effects were observed for adolescents with higher pretest test anxiety and socially prescribed perfectionism. Our findings demonstrate that even a short program can yield positive effects, particularly for adolescents with high program attendance and who experience high problem severity and maladaptive perfectionism.

## 1. Introduction

Adolescence is a period of remarkable transformations, characterized by the development of identity and social skills, a heightened sensitivity to stress, and an increased risk of developing mental health problems ([1]; [4]; [40]). In addition, in today’s society, adolescents frequently experience elevated levels of stress and performance anxiety due to high expectations from themselves and others ([57]). This contributes to their vulnerability of experiencing mental health issues ([25]), such as internalizing and externalizing problems and reduced wellbeing ([7]; [42]). Among the most salient negative stressors experienced by adolescents are stressors related to school, such as taking exams, keeping up with schoolwork, and failure in exams ([2]; [35]). In the Netherlands, 27–47% of secondary school students experience school-related stress and pressure, and almost one in three secondary school students regularly or often experience pressure to meet their own or other people’s expectations ([3]; [26]). Moreover, in the last two decades, there have been increases in school stress and pressure in higher-income European countries ([3]; [11], [12]; [45]), as well as higher expectations from oneself or others ([14]), enlarging the need for supporting adolescents with school-related stress and performance anxiety.

In an effort to reach adolescents who can benefit from programs preventing (the escalation of) mental health problems, school-based intervention programs have been developed, including programs targeting stress management, social skills, and test anxiety. Systematic and meta-analytic reviews indicate moderate effectiveness of such school-based programs overall ([15]; [17]; [43]; [52]); however, there are inconsistent effects across individual studies. In fact, meta-analyses demonstrate substantial heterogeneity in effects across studies, to which various factors seem to contribute, including the target group ([52]), dosage ([15]), and level of implementation ([17]). Yet, individual studies examining the effectiveness of school-based programs often focus on the overall effectiveness rather than on moderators of effectiveness. This precludes obtaining an overview of who benefitted from the program and under what circumstances. Insight into the moderators of effectiveness of school-based intervention programs is important for various reasons ([5]). First, it can identify (groups of) individuals that are most likely or unlikely to benefit from a program, guiding which programs should be offered to specific groups. Second, it could help the development and implementation of intervention programs by revealing factors crucial for effectiveness, such as greater program adherence or attendance, which has previously been related to greater program gains ([58]). Finding no significant moderators of effectiveness is also informative, as this demonstrates that programs are effective for a diverse and broad group of individuals ([5]).

The current study aims to provide insight into the moderators of effectiveness of a school-based skills-training program for adolescents addressing performance anxiety, encompassing test anxiety, and fear of failure. As demonstrated with a randomized controlled trial (RCT), this program had a small overall effect on reducing adolescents’ test anxiety. Furthermore, for adolescents with high program attendance, i.e., adolescents who attended more than half of the sessions, the program had small effects on reducing test anxiety and fear of failure ([53]). The present study aims to extend our understanding of the effectiveness of this program by investigating the variation in program effectiveness across individuals. More specifically, this study investigated if program effectiveness is moderated by higher problem severity (i.e., higher pretest levels of performance anxiety), as well as higher expectations from oneself or from others, manifested as maladaptive perfectionism and parental pressure, and whether this depends on the level of program attendance.

Based on previous meta-analyses ([5]; [24]; [28]; [47]; [46]; [52]), we hypothesized the effects of the performance anxiety program to be larger for adolescents with higher pretest performance anxiety (i.e., fear of failure and test anxiety). Because of high severity of problems at baseline, there is supposed to be more room for improvement and, thus, larger program effects in this group ([47]; [46]). In addition, more severe symptoms or problems can result in more motivation for change or to learn (new) skills to deal with these symptoms ([47]; [46]), as the participants are likely to experience problems related to their performance anxiety.

Perfectionism is considered to be a multidimensional cognitive process, defined as striving for flawlessness and the pursuit of excessively high standards for performance, combined with critical self-evaluation ([21]). Perfectionism can be conceptualized at both an intrapersonal and interpersonal levels, with several dimensions. These include self-oriented perfectionism (SOP), that is, perfectionistic standards directed by oneself and toward oneself (i.e., intrapersonal level), and socially prescribed perfectionism (SPP), which is the belief that others hold perfectionistic expectations and motives for oneself (i.e., interpersonal level; [21]). Additionally, SOP can be divided into SOP striving, that is, striving for high standards, and SOP critical, that is, self-criticism when one does not attain perfection and has excessive concern over mistakes ([21]). Socially prescribed perfectionism and self-criticism are considered as maladaptive, and therefore, the focus of this study, while striving for high personal standards, is considered an adaptive, healthy trait ([20]; [37]).

Several features of maladaptive perfectionism, such as procrastination, social disconnection (e.g., heightened rejection sensitivity; a view of the self as irrelevant to others), and high self-presentation (e.g., the need to appear perfect; the need to portray a flawless image) ([20]; [22]; [23]), might make it more difficult to benefit from interventions. For example, individuals who are hyper-vigilant to signs of rejection or have the tendency to try to cover up their perceived shortcomings might be hesitant to disclose important (intimate) information to a therapist or instructor, which may create problems with establishing a good working alliance with the therapist, negatively affecting the program outcome ([20]; [23]; [29]; [59]). Furthermore, individuals might procrastinate to perform a task or homework to avoid less-than-perfect performance ([22]), hindering program success. Previous reviews suggest that maladaptive perfectionism has a negative impact on program processes and outcomes ([31]; [41]). To illustrate, one RCT of a group-based cognitive behavioral treatment (CBT) program for children at risk for anxiety and/or depression (8–11 years) showed that pre-treatment self-criticism was related to higher post-treatment depression scores ([32]). Another RCT of a group-based CBT program for clinically anxious children (6–13 years) demonstrated that self-oriented perfectionism predicted higher levels of anxiety symptoms ([30]), whereas socially prescribed perfectionism was not related to depression or anxiety ([30]; [32]). Yet another study of a universal school-based cognitive behavior prevention program for anxiety in children (9–12 years) revealed that high levels of perfectionism (both SOP and SPP) were associated with increased pre- to postintervention anxiety levels ([18]). Yet, these studies did not assess if perfectionism moderated program effectiveness. As maladaptive perfectionism is related with higher levels of mental health problems ([21]; [27]; [37]), leaving more room for improvement or more motivation to change, it may also be that maladaptive perfectionism is associated with higher program effectiveness. Taken together, maladaptive perfectionism may either hinder or promote program effectiveness. As research on its role as a moderator is lacking, it is important for program development and implementation to clarify how maladaptive perfectionism impacts program effectiveness. Furthermore, investigating both the intrapersonal and interpersonal dimension of maladaptive perfectionism may clarify what aspect of maladaptive perfectionism is most relevant to program effectiveness and should, therefore, be taken into account. This might especially be important in adolescents, as adolescence is a developmental phase characterized by the formation of identity, seeking independence from parents and increasing peer influence ([4]; [6]).

Another potential moderator of program effectiveness is parental pressure, referring to the perception of high parental expectations and excessive parental criticism ([22]). Self-evaluations of performance are tied to assumptions about parental expectations, and failure to meet these expectations may lead to disapproval or potential loss of love and acceptance ([22]). Previous RCT research showed that some parenting behaviors, such as negative affect, low emotional warmth, and overinvolvement, are associated with less favorable CBT program outcomes in clinically anxious youth (6–18 years) ([13]; [19]). These negative parenting styles might limit youth to develop coping skills and self-competence to deal with challenges, hindering program success ([19]). Therefore, it is expected that adolescents with higher perceived parental pressure benefit less from the performance anxiety program.

In sum, it is crucial to better understand individual characteristics that affect program effectiveness to optimize successful implementation and the tailoring and optimalization of programs to meet the specific needs of different (groups of) participants. The current study aims to identify whether problem severity (i.e., pretest levels of performance anxiety), maladaptive perfectionism (i.e., self-criticism and socially prescribed perfectionism), and perceived parental pressure moderate the effectiveness of a school-based skills-training program addressing skills to deal with performance anxiety and whether this depends on the level of program attendance. It is expected that adolescents with higher problem severity benefit more from the program. No clear hypotheses are formulated for maladaptive perfectionism, as the previous literature is inconsistent. Finally, it is expected that adolescents with higher parental pressure benefit less from the program. If some characteristics promote or hinder program effectiveness for adolescents in need, programs should be adapted to address the unique needs of these (groups of) individuals. Overall, identifying factors associated with program gains or failure may improve successful implementation.

## 2. Methods

### 2.1. Design and Procedure

Implementation and data collection occurred at nine secondary schools in one of the four largest city in the Netherlands between 2018 and 2021 ([53]). After a three-session psychoeducation program about stress ([56]), the students could self-select into one of the skills-training programs offered at school, either a performance anxiety program or a social skills-training program ([53]). The current study only focuses on the performance anxiety program. An RCT was conducted for a school-based skills-training program addressing skills to deal with performance anxiety ([53]).

The students and parents received written information about the programs and research and provided active informed consent. The students were randomly allocated into the experimental or waitlist control group. Before (T1) and after (T2) the implementation of the skills-training programs, program outcomes were assessed for both groups. The present study is registered in the International Clinical Trials Registry Platform (Netherlands Trial Register, number NTR7680), is published as a study protocol ([54]), and the design is approved by the Ethical Committee Psychology of Leiden University (CEP18-1105/419; first submission was approved on 4 December 2018; two amendments have been approved at a later stage). The data collection started in February 2019. Overall effectiveness results are presented elsewhere ([53]).

### 2.2. Participants

A total of 211 students enrolled in the performance anxiety program, with *N* = 104 in the experimental group and *N* = 107 in the control group. A total of 15 participants dropped out of the study at T2 and were excluded from the analyses (*N* = 9 in the experimental group; *N* = 6 in the control group). Hence, the final sample consisted of *N* = 196 participants. The students were between 12 and 17 years old (*M_age_* = 14.12, *SD* = 0.79, 53% females). At T1, the participants were in the first (15%), second (63%), third (13%), or fourth year (9%) of secondary school. The participants were practical–prevocational education students (33%), prevocational–senior general education students (28%), or senior general–preuniversity education students (39%). More than half of the participants had a Western ethnic identity (56%, e.g., Dutch); other participants had a mixed (Western–non-Western, 13%, e.g., Dutch–Turkish) or non-Western (31%, e.g., Moroccan) ethnic identity. Most participants were born in the Netherlands (84%) and lived with both of their parents (66%). The sample was representative for Dutch adolescents in terms of educational level and minority background ([53]).

### 2.3. Performance Anxiety Program

The performance anxiety programs were provided at school by trained professionals from three youth care organizations who had at least one year of experience with being a trainer and completed at least higher vocational education. The program, adjusted from existing programs, consisted of seven weekly sessions and was designed to fit the schedule of the schools (so a session could fit in one lesson of 45 min). The program consisted of psychoeducation and teaching cognitive coping strategies to deal with (school) pressure, in addition to relaxation techniques. The control group did not receive any training during the implementation of the skills-training programs in the experimental group. They received the program after completion of the assessment at T2 (approximately 8 weeks after the experimental group started the skills-training program). Regarding program integrity, on average, 90% of program assignments were implemented ([53]).

### 2.4. Instruments

#### 2.4.1. Performance Anxiety

*Test anxiety* was assessed with the Spielberger Test Anxiety Inventory (TAI) ([44]; [51]). The TAI consists of 20 items (e.g., “During tests I feel very tense”) rated on a 4-point scale ranging from 1 (*almost never*) to 4 (*almost always*). Previous research showed adequate validity and internal consistency ([51]). Mean scores were computed, with higher scores reflecting more test anxiety. In the current sample, internal consistency is high at both timepoints (α = 0.93 at T1 and α = 0.89 at T2).

*Fear of failure* was assessed with the short form of the Performance Failure Appraisal Inventory (PFAI; [9]). The PFAI consists of 5 items (e.g., “When I am failing, I am afraid that I might not have enough talent”) measured on a 5-point scale ranging from 1 (*do not believe at all*) to 5 (*believe 100% of the time*). Previous research showed adequate validity and internal consistency ([9]). Mean scores were computed, with higher scores reflecting more fear of failure. In the current sample, internal consistency is high at both timepoints (α = 0.83 at T1 and α = 0.88 at T2).

#### 2.4.2. Moderators

*Problem severity* was defined as the pretest (T1) level of performance anxiety, distinguishing the pretest level of test anxiety and pretest level of fear of failure.

*Maladaptive perfectionism* was assessed at T1 with the Child and Adolescent Perfectionism Scale (CAPS-14; [36]), translated into Dutch. The instrument consists of 14 items rated on a 5-point Likert scale ranging from 1 (*false*) to 5 (*true*). Two subscales of perfectionism were used: self-oriented critical perfectionism or self-criticism (SOP-C, four items; “I get mad at myself when I make a mistake”) and socially prescribed perfectionism (SPP, seven items; “Other people always expect me to be perfect”). Previous research showed good internal consistency ([36]). Mean scores were computed, with higher scores reflecting more maladaptive perfectionism. In the current sample, internal consistency is high for both subscales (α = 0.88 for SOP-C and α = 0.82 for SPP).

*Perceived parental pressure* was measured at T2 with the Multidimensional Inventory of Perfectionism in Sport (MIPS) with the subscale for perceived parental pressure ([48]). The subscale consists of eight items, including parental expectations (e.g., “My parents set extremely high standards for me”) and criticism (e.g., “My parents criticize everything I do not do perfectly”), rated on a 6-point scale of 1 (*never*) to 6 (*always*). Previous studies showed high internal consistency (above 0.92, [48]; [49]). Mean scores were computed, with higher scores presenting more perceived parental pressure. The current sample has an excellent internal consistency of 0.94.

### 2.5. Data Analysis

Statistical analyses were performed with SPSS version 29. First, descriptive and correlation analyses (Pearson and point-biserial correlations) were performed for all study variables. Second, separate univariate analysis of covariance (ANCOVA) tests were conducted to determine the influence of the moderators (i.e., problem severity, self-criticism, socially prescribed perfectionism, and parental pressure) on the two indicators of performance anxiety, that is, test anxiety and fear of failure (*N* = 8 models). The models were tested in the full sample, using an intention-to-treat approach, as well as in a subsample of adolescents with higher program attendance (i.e., adolescents who attended four or more sessions (*N* = 46) ([53]). The post-test outcome measures (T2) were included as dependent variables, and the condition (i.e., experimental and control groups) was included as a fixed factor. The pretest outcome measures (T1) and gender were included as covariates in the analyses, as there were more females in the experimental group than in the control group ([53]). The moderators and interaction terms (moderator * condition) were added as additional factors to the ANCOVAs. In the models with problem severity (i.e., pretest test anxiety or fear of failure) as the moderator, only the interaction term was added. The moderators were centered around the mean. To interpret significant moderator effects, we divided the sample based on the median of the moderator variable and conducted separate ANCOVA tests on each subgroup. Cohen’s *d*s were calculated using an online effect size calculator ([39]) based on the η^2^. Small effect sizes were considered as *d* = 0.20, moderate effect sizes as *d* = 0.50, and large effect sizes as *d* = 0.80 ([8]).

## 3. Results

### 3.1. Full Sample

Table 1 presents the means, *SD*s, and correlations for the study variables. Table 2 demonstrates the results of the moderator analyses in the full sample. For test anxiety, a small significant moderating effect was found for problem severity (*F*(1, 192) = 7.02; *p* = 0.01; *d* = 0.38). For participants with low levels of baseline test anxiety, the program was not effective (*F*(1, 97) = 0.34; *p* = 0.56), while participants with high levels of baseline test anxiety benefitted from the program (*F*(1, 91) = 8.20; *p* = 0.01; *d* = 0.60). Figure 1 demonstrates the moderating effect of problem severity. For test anxiety, self-criticism, socially prescribed perfectionism, and perceived parental pressure did not moderate program effectiveness. For fear of failure, problem severity, self-criticism, socially prescribed perfectionism, and perceived parental pressure did not moderate program effectiveness.

### 3.2. Subsample with Higher Program Attendance

Table 3 presents the results of the moderator analyses for participants with higher program attendance. For test anxiety, small significant moderating effects were found for problem severity (*F*(1, 142) = 6.48; *p* = 0.01; *d* = 0.43), self-criticism (*F*(1, 140) = 4.04; *p* = 0.05; *d* = 0.34), and socially prescribed perfectionism (*F*(1, 140) = 6.15; *p* = 0.01; *d* = 0.42). For problem severity, separate ANCOVAs in the subsamples with low (*F*(1, 68) = 2.71; *p* = 0.10) and high levels of baseline test anxiety (*F*(1, 69) = 0.52; *p* = 0.48) did not reveal program effectiveness in either of the samples. Even though there was no significant post hoc effect, the significant moderating effect of problem severity seems to be driven by adolescents with high baseline test anxiety. The findings demonstrated that the post-test levels in the experimental group were reduced for participants with high baseline test anxiety (T1: *M* = 2.61, *SD* = 0.42; T2: *M* = 2.28, *SD* = 0.54) while being slightly increased for participants with low baseline test anxiety (T1: *M* = 1.59, *SD* = 0.29; T2: *M* = 1.71, *SD* = 0.32; see Figure 2).

For self-criticism, the performance anxiety program resulted in a moderate reduction in test anxiety for participants with high levels of SOP-C (*F*(1, 73) = 5.03; *p* = 0.03; *d* = 0.52), whereas the program was not effective for participants with low levels of SOP-C (*F*(1, 61) = 0.17; *p* = 0.69). Figure 3 demonstrates the moderator effect for self-criticism. For socially prescribed perfectionism, there were no significant differences between the experimental and control group in the group of participants with low (*F*(1, 69) = 0.53; *p* = 0.47) and high levels of SPP (*F*(1, 65) = 0.47; *p* = 0.50). Even though there was no significant post hod effect, the significant moderating effect of SPP seem to be driven by adolescents with high levels of SPP, as the findings demonstrated that test anxiety was reduced for participants with high levels of SPP who attended the program (T1: *M* = 2.47, *SD* = 0.65; T2: *M* = 2.09, *SD* = 0.59) while staying relatively the same for participants with low levels of SPP (T1: *M* = 2.09, *SD* = 0.49; T2: *M* = 2.11, *SD* = 0.50) (see Figure 4).

For fear of failure, no moderator effects were found for problem severity, self-criticism, socially prescribed perfectionism, and perceived parental pressure.

## 4. Discussion

Given the increases in school stress and pressure over the last two decades ([3]; [11], [12]; [45]), which are associated with negative mental health outcomes, it is necessary to support adolescents with school-related stress and performance anxiety, manifested as the fear of failure and test anxiety. Moreover, it is crucial for tailoring and optimizing of intervention programs to better understand which individuals are most likely or unlikely to benefit from such programs and which circumstances affect program effectiveness. Therefore, the current study aimed to identify whether problem severity, maladaptive perfectionism, and perceived parental pressure moderated the effectiveness of a school-based program addressing skills to deal with performance anxiety and whether the level of program attendance played a role. The results showed that for test anxiety, the program was only effective for adolescents with higher pretest levels. Furthermore, when program attendance was high, the program was only effective for adolescents with higher self-criticism, and larger effects were observed for adolescents with higher problem severity and socially prescribed perfectionism. Perceived parental pressure did not moderate program effectiveness and there were no significant moderator effects for fear of failure. Interestingly, moderation effects were only observed for test anxiety, not for the broader, more general form of performance anxiety: fear of failure. Test anxiety is school specific, while fear of failure is a broader construct and may manifest in various contexts beyond the academic environment, for instance, in sports, hobbies, or personal relationships. It is possible that the observed difference stems from the specific focus of the program and the immediate applicability to the academic context. The performance anxiety program focused on school-related stress and related coping skills and addressed scenarios related to the school context that are closely linked to (the measurements of) test anxiety and maladaptive perfectionism.

The finding that particularly adolescents with higher baseline test anxiety benefitted from the program is in line with previous research in adolescents ([5]; [24]; [28]; [47]; [46]; [52]), highlighting that such programs have the potential to support adolescents who are at high risk or express a need for support in dealing with performance anxiety. Adolescents with higher problem severity are likely to be more motivated to engage in the program and have more to gain, which yields greater program benefits ([47]; [46]). This suggests that these adolescents should be targeted for selective or indicated intervention programs to promote adolescent mental health. Moreover, it also advocates the use of screening methods at recruitment to select adolescents with high problem severity at baseline.

Most effects were found in the subsample of adolescents who attended more than half of the program. Unsurprisingly, these adolescents reported more needs at pretest (see Appendix A), confirming higher program engagement for those with higher problem severity and maladaptive perfectionism. At higher levels of program attendance and, therefore, more exposure to the program, it becomes apparent that adolescents with maladaptive perfectionism benefit (more) from the program. As maladaptive perfectionistic tendencies, such as procrastination, social disconnection, and a need to appear perfect might hinder youth to benefit from interventions, higher attendance allows for repeated exposure to coping strategies, more opportunities to practices these skills, and more time to build trust with the trainer(s), normalize experiences, and learn from peers. Alternatively, larger program effectiveness for adolescents with maladaptive perfectionism in this subsample might be related to higher problem severity, as these adolescents reported higher performance anxiety at baseline.

Our results indicate that higher maladaptive perfectionism, that is, self-criticism and socially prescribed perfectionism, is associated with larger program effects. This is in contrast with theoretical evidence that suggested that maladaptive perfectionism was related to less beneficial effects. These contrasting results might be related to the discussion of mostly (sub)clinical samples. Potentially, perfectionistic features may be more problematic in (sub)clinical samples, as these coincide with other mental health issues (e.g., obsessive-compulsive disorder, anxiety, and depression) and additional problematic behaviors, interfering with treatment and skill learning. For instance, clinical college student samples have reported higher levels of maladaptive perfectionism compared to non-clinical samples ([33]). It is possible that in our sample, the level of perfectionism was not extreme enough to prevent benefitting from the program. The focus of our performance anxiety program was on adolescents struggling with dealing with these problems, and since maladaptive perfectionism is strongly related to the fear of failure and test anxiety and may share some underlying mechanisms ([10]; [50]), it is likely that the program worked better for adolescents with high levels of maladaptive perfectionism. Moreover, since maladaptive perfectionism is positively related with perceived stress and mental health problems ([21]; [27]; [37]), it is possible that adolescents experiencing more problems at baseline are more motivated to engage in the program and have more to gain, yielding larger program effects ([47]; [46]). Our results support this, as experiencing more test anxiety at baseline was positively associated with maladaptive perfectionism. These findings indicate that school-based performance anxiety programs are also beneficial for adolescents with high levels of maladaptive perfectionism. Both the intrapersonal (i.e., self-criticism) and interpersonal (i.e., socially prescribed perfectionism) dimensions of perfectionism seem to be relevant to program effectiveness.

Perceived parental pressure—consisting of high parental expectations and excessive parental criticism—did not influence program effectiveness, which contradicts earlier studies showing that negative parenting styles were associated with less favorable program outcomes in anxious youth ([13]; [19]). However, these studies were conducted in younger samples and in youth with anxiety disorders ([13]; [19]). First, it is possible that perceived parental pressure did not moderate program effectiveness in this age group, because adolescents have greater independence from parents than younger adolescents or children, which might make adolescents less susceptible to parental pressure. Additionally, this elevated independence in adolescence may create less opportunity for parents to impose high expectations and criticize their children. Indeed, recent research showed that perceived parental pressure was lower in older than younger athletes ([16]). Second, it is likely that in clinical samples, problems are more complex or severe, making parents more involved and potentially exerting more pressure, hindering treatment effectiveness. Indeed, a previous study demonstrated that children with generalized anxiety disorder reported high levels of overprotection, parental pressure, and acute life events compared to non-patient school controls ([34]). Interestingly, socially prescribed perfectionism, i.e., the belief that others—not only parents—hold perfectionistic expectations, did affect program effectiveness. Possibly, adolescents’ sensitivity to peer norms might be a crucial factor to explain these divergent findings. As peers and friends become more important during adolescence, peer influence and pressure could have a significant role in adolescents’ attitudes and behavior ([6]) and, therefore, also in program effectiveness. Future research in this area is warranted. Even though our findings demonstrated that parental pressure did not influence program effectiveness, it does not mean that parents have no influence at all. Previous research showed that some parenting styles, such as negative affect and overinvolvement, influenced program outcomes ([13]; [19]). Hence, future research should investigate different forms of parental influence on program effectiveness, including different parenting styles such as overprotection and behavioral control.

### Strengths and Limitations

Despite the diverse and representative sample of Dutch adolescents in our study, with adolescents from various cultural backgrounds and educational levels, program implementation was disrupted by the COVID-19 pandemic ([53]). Some skills-training programs were postponed and restarted, which resulted in a lower program attendance and might have affected program implementation and outcome. However, program enrollment before or after the COVID-19 pandemic did not influence program effectiveness, suggesting that the COVID-19 pandemic had little impact ([53]).

We used a waitlist control design, so long-term follow-up was not possible. Future research should include follow-up assessments, as larger effects were found for follow-up compared to postintervention assessments for stress reduction interventions ([52]), and program gains were only found at follow-up for young adolescents after a cognitive behavioral prevention program ([18]). It might be that adolescents need to practice their newly learned skills in real-life settings in order for the program effects to unfold.

Another methodological limitation is the use of some of the questionnaires. We used the 14-item version of the CAPS ([36]), based on the original version ([21]), which has been shown to have acceptable reliability for use in general research for SPP, while SOP-C was only recommended for explorative research ([55]). Nevertheless, in our sample the reliability was good for both maladaptive perfectionism subscales (>0.80). Furthermore, we measured perceived parental pressure at T2, which could have influenced the results. It is possible that parental pressure changed over time (from baseline to T2) as a result of the program. However, we expected that perceived parental pressure remained stable during the course of our study, as it has high temporal stability ([38]), and parents were not actively involved in the program.

## 5. Conclusions and Implications

In conclusion, the current study examined moderators of effectiveness for a school-based performance anxiety program. Our findings demonstrate that even a short program of seven weekly 45 min sessions can reduce test anxiety in adolescents with higher problem severity and maladaptive perfectionism, especially when program attendance is high. As such, schools should be aware that low-threshold school-based programs have the potential to support adolescents in need, making it worthwhile to consider implementing such programs in the school setting. To optimize results, efforts should be made to motivate adolescents to attend and engage in such school-based intervention programs, as this allows them to benefit more.

## Figures and Tables

**Figure 1 behavsci-15-00436-f001:**
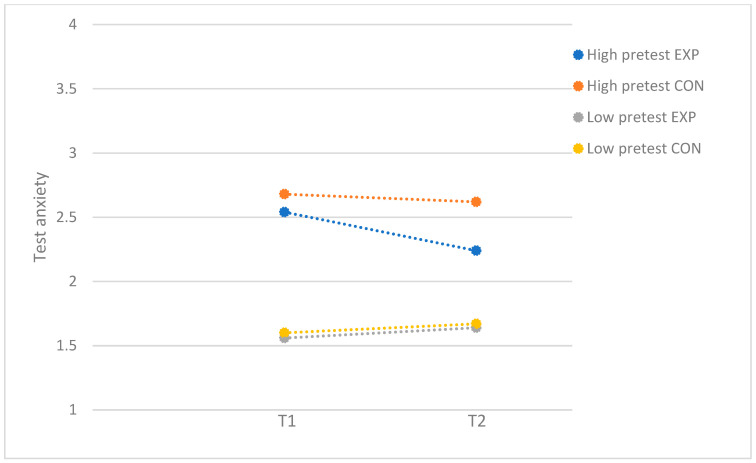
Test anxiety: moderation by problem severity, with full sample. EXP = experimental group; CON = control group.

**Figure 2 behavsci-15-00436-f002:**
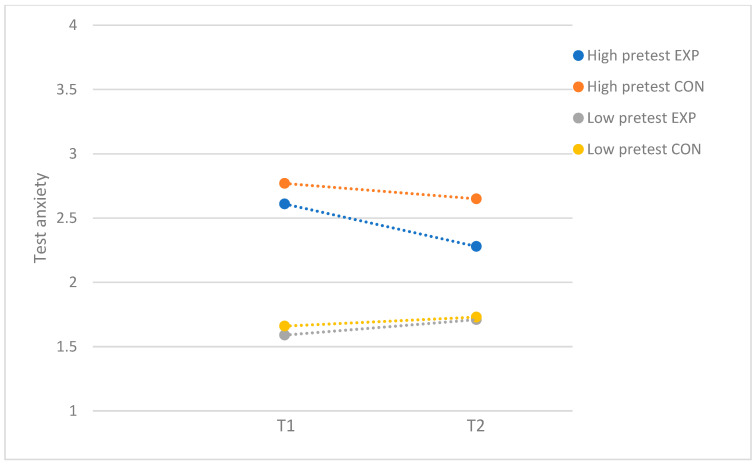
Test anxiety: moderation by problem severity, with subsample of participants who attended more than half of the sessions. EXP = experimental group; CON = control group.

**Figure 3 behavsci-15-00436-f003:**
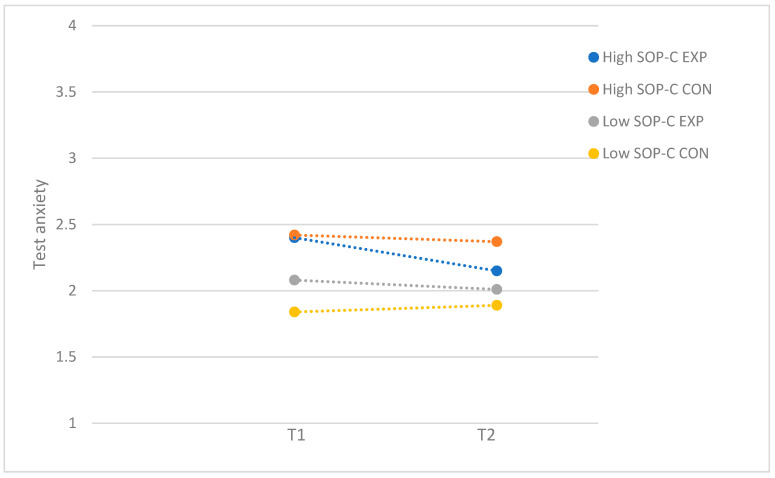
Test anxiety: moderation by self-criticism perfectionism (SOP-C), with subsample of participants who attended more than half of the sessions. EXP = experimental group; CON = control group.

**Figure 4 behavsci-15-00436-f004:**
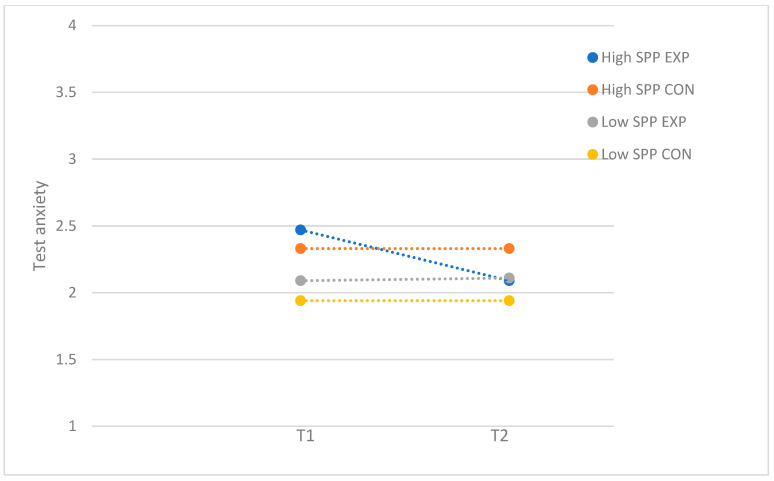
Test anxiety: moderation by socially prescribed perfectionism (SPP), with subsample of participants who attended more than half of the sessions. EXP = experimental group; CON = control group.

**Table 1 behavsci-15-00436-t001:** Correlations for the study variables of the full sample.

	1 ^a^	2	3	4	5	6	7	*M*	*SD*	Range
1. Gender ^b^	-							-	-	-
2. Test anxiety (T1)	0.15 *	-						2.08	0.62	1.00–3.90
3. Test anxiety (T2)	0.11	0.78 ***	-					2.03	0.59	1.00–3.55
4. Fear of failure (T1)	0.23 **	0.65 ***	0.53 ***	-				2.16	0.91	1.00–4.40
5. Fear of failure (T2)	0.26 ***	0.60 ***	0.67 ***	0.62 ***	-			2.20	0.94	1.00–4.80
6. Self-oriented perfectionism-critical (T1)	0.08	0.51 ***	0.34 ***	0.49 ***	0.42 ***	-		2.43	0.99	1.00–5.00
7. Socially prescribed perfectionism (T1)	−0.02	0.42 ***	0.33 ***	0.37 ***	0.28 ***	0.43 ***	-	2.59	0.93	1.00–4.71
8. Perceived parental pressure (T2)	−0.10	0.24 ***	0.36 ***	0.22 **	0.26 ***	0.38 ***	0.50 ***	2.34	1.11	1.00–6.00

^a^: Point-biserial correlations were performed. ^b^: 0 = male adolescents; 1 = female adolescents. *** = *p* < 0.001; ** = *p* < 0.01; * = *p* < 0.05.

**Table 2 behavsci-15-00436-t002:** Results of the moderator analyses for test anxiety and fear of failure.

	Test Anxiety (T2)		Fear of Failure (T2)	
	*F* (*p*)	η^2^	*F* (*p*)	η^2^
Problem severity (pretest levels; T1)	**7.02 (0.01)**	0.04	0.36 (0.55)	0.00
Self-oriented perfectionism-criticism (SOP-C; T1)	0.85 (0.36)	0.00	0.30 (0.58)	0.00
Socially prescribed perfectionism (SPP; T1)	2.80 (0.10)	0.02	0.49 (0.49)	0.00
Perceived parental pressure (T2)	1.14 (0.29)	0.01	0.86 (0.36)	0.01

Note: significant results are in bold.

**Table 3 behavsci-15-00436-t003:** Results of the moderator analyses for test anxiety and fear of failure for participants who attended four or more sessions.

	Test Anxiety (T2)		Fear of Failure (T2)	
	*F* (*p*)	η^2^	*F* (*p*)	η^2^
Problem severity (pretest levels; T1)	**6.48 (0.01)**	0.04	2.19 (0.14)	0.02
Self-oriented perfectionism-criticism (SOP-C; T1)	**4.04 (0.05)**	0.03	0.00 (0.98)	0.00
Socially prescribed perfectionism (SPP; T1)	**6.15 (0.01)**	0.04	0.09 (0.77)	0.00
Perceived parental pressure (T2)	0.91 (0.34)	0.01	1.23 (0.27)	0.01

Note: significant results are in bold.

## Data Availability

The datasets generated and analyzed during the current study are not publicly available but are available from the corresponding author on request.

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
