# Peer review of "What Works for Whom? The Influence of Problem Severity, Maladaptive Perfectionism, and Perceived Parental Pressure on the Effectiveness of a School-Based Performance Anxiety Program"

_behavsci, 2025, doi:10.3390/bs15040436_

Round 1
Reviewer 1 Report
Comments and Suggestions for Authors
The authors set out to investigate the effectiveness of a short (3+7 weeks) school-based program for adolescents testing the contribution of moderators and participation. The present manuscript is a continuation of another paper in which the overall effectiveness of the program was analysed.
The study is correctly designed and described, the manuscript is well written. However, there is one major problem with the description of variables, namely the construct of “problem severity” and its assessment cannot be deciphered from the text. Description in the Introduction (p4L152) hints at “problem severity” referring to pretest levels of performance anxiety but nowhere in Methods is it stated how it was exactly assessed. Without description, the reader cannot understand how T1 measurements of “test anxiety” and “fear of failure” were contracted into one variable (if that is how “problem severity” was assessed), or the results in Table2 can also not be comprehended. The time of measurement for variables in Table2 should also be specified.
Unsurprisingly, the authors found higher effects for those adolescents who attended more than half of the program. Therefore, it is reasonable to suppose that differences in T1 measurements of performance anxiety and moderating variables would be found between adolescents who attended more than half of the programs versus those who did not. The authors should insert such a table which might be useful for identifying those features which make it most likely for students to complete the program before its start.
The reasoning for choosing line diagrams in Figures 1-4 can be understood but these are misleading since data were measured only in 2 timepoints. A point diagram connecting points with dotted lines could better represent this type of measurement.
The authors need to correct “program severity” to “problem severity” on p6 L245, p6 L273, p10L319, p11L329.
The citations and reference list conform to the requirements of the journal (“A reference list is always alphabetically arranged”) but references are also numbered which is unnecessary and misleading.
The Harvard system cites references by authors-year and produces an alphabetical reference list by author name. The Vancouver system cites references numbered and produces a reference list by citation numbers in increasing order. The two should not be mixed.
Author Response
The authors set out to investigate the effectiveness of a short (3+7 weeks) school-based program for adolescents testing the contribution of moderators and participation. The present manuscript is a continuation of another paper in which the overall effectiveness of the program was analysed.
- The study is correctly designed and described, the manuscript is well written. However, there is one major problem with the description of variables, namely the construct of “problem severity” and its assessment cannot be deciphered from the text. Description in the Introduction (p4L152) hints at “problem severity” referring to pretest levels of performance anxiety but nowhere in Methods is it stated how it was exactly assessed. Without description, the reader cannot understand how T1 measurements of “test anxiety” and “fear of failure” were contracted into one variable (if that is how “problem severity” was assessed), or the results in Table2 can also not be comprehended. The time of measurement for variables in Table2 should also be specified.
We appreciate the positive remarks on our study. In the manuscript, we used the term “problem severity” instead of “higher pretest/baseline levels of performance anxiety” to avoid long, complicated sentences. To clarify this, we added an explanation the first time problem severity is mentioned in the introduction (page 4; “i.e., higher pretest levels of performance anxiety”), as well as an extra paragraph in the methods section (page 9, under 2.4.2. moderators; “Problem severity was defined as the pretest (T1) level of performance anxiety, distinguishing pretest level of test anxiety and pretest level of fear of failure.” and under 2.5 Data Analyses (page 11): “pretest test anxiety or fear of failure”. Furthermore, measurement timepoints were added to the tables (Tables 1, 2, 3 and 4).
- Unsurprisingly, the authors found higher effects for those adolescents who attended more than half of the program. Therefore, it is reasonable to suppose that differences in T1 measurements of performance anxiety and moderating variables would be found between adolescents who attended more than half of the programs versus those who did not. The authors should insert such a table which might be useful for identifying those features which make it most likely for students to complete the program before its start.
Thank you for this insightful comment. We included a table in the Supplementary Material (Table S1) with means and SD’s of the two subgroups of the experimental group (adolescents who attended less than four sessions and adolescents who attended four or more sessions), including differences between the groups. Indeed, adolescents with higher program attendance had higher levels of performance anxiety and maladaptive perfectionism before the start of the program, indicating that adolescents who needed it the most attended more sessions of the performance anxiety program. We added this in the discussion section (page 17): “Unsurprisingly, these adolescents reported more needs at pretest (see Supplementary Table S1), confirming higher program engagement for those with higher problem severity and maladaptive perfectionism.”.
- The reasoning for choosing line diagrams in Figures 1-4 can be understood but these are misleading since data were measured only in 2 timepoints. A point diagram connecting points with dotted lines could better represent this type of measurement.
We changed the figures into a plot with connecting points and dotted lines instead of line diagrams (Figures 1-4).
- The authors need to correct “program severity” to “problem severity” on p6 L245, p6 L273, p10L319, p11L329.
We corrected these typos.
- The citations and reference list conform to the requirements of the journal (“A reference list is always alphabetically arranged”) but references are also numbered which is unnecessary and misleading.
The Harvard system cites references by authors-year and produces an alphabetical reference list by author name. The Vancouver system cites references numbered and produces a reference list by citation numbers in increasing order. The two should not be mixed.
We used a citation format that arranged the references in alphabetic order of the first author (APA, 7th addition). Numbers were added in the review process.
Reviewer 2 Report
Comments and Suggestions for Authors
Thank you for the opportunity to review the manuscript “What Works for Whom? The Influence of Problem Severity, Maladaptive Perfectionism and Perceived Parental Pressure on the Effectiveness of a School-Based Performance Anxiety Program”. Overall I found this paper with good writing quality, and sufficient discussions of the necessity to address the research gap. I have a few concerns that prevent this paper to be published in its current version, which I detail below.
- In the Introduction section, more rationale of SOP-C and SPP is needed. The authors had one statement indicating perfectionism can be conceptualized at both inter- and intra-personal level, but it is not clear how the examined sub-dimensions of perfectionism fit into these levels, and if authors anticipate the same moderating role of SOP-C and SPP, it is not clear why these two constructs cannot be combined into one composite of perfectionism. Similarly, the Discussion section merges the roles of SOP-C and SPP together without interpreting whether and why these two moderators function same or different in the examined program efficacy from a theoretically perspective.
- Clear hypotheses are needed at the end of the Introduction session. The current writing is very vague and it is not clear what findings are anticipated from the study.
- I appreciate the authors’ efforts in including gender as a potential covariate. I am wondering whether other covariates should be considered as well. For example, could students’ grade level be related to program efficacy as students may have different course/test loads across different school grades?
- There are a few typos in the manuscript, for example, a few places listed “program severity”.
- I was wondering whether the data captures any of students’ peer contexts. Adolescence is the developmental stage when peers and friends become important socialization agents and peer influence and pressure could have outstanding role in adolescents’ emotions and behaviors. Given the current study does not find parental pressure as a moderator, peer context might be something to consider.
- In Section 4.1, I would suggest revise the leading sentence to be more cautious. Although this study includes culturally diverse participants, I’m worried the sample size cannot be treated as large with ~120 participants.
I hope the authors find my comments helpful.
Author Response
Thank you for the opportunity to review the manuscript “What Works for Whom? The Influence of Problem Severity, Maladaptive Perfectionism and Perceived Parental Pressure on the Effectiveness of a School-Based Performance Anxiety Program”. Overall I found this paper with good writing quality, and sufficient discussions of the necessity to address the research gap. I have a few concerns that prevent this paper to be published in its current version, which I detail below.
- In the Introduction section, more rationale of SOP-C and SPP is needed. The authors had one statement indicating perfectionism can be conceptualized at both inter- and intra-personal level, but it is not clear how the examined sub-dimensions of perfectionism fit into these levels, and if authors anticipate the same moderating role of SOP-C and SPP, it is not clear why these two constructs cannot be combined into one composite of perfectionism. Similarly, the Discussion section merges the roles of SOP-C and SPP together without interpreting whether and why these two moderators function same or different in the examined program efficacy from a theoretically perspective.
We agree with the reviewer that it would be helpful to add more rationale of SOP-C and SPP. We added information that SOP fits into the intrapersonal level, and SPP in the interpersonal level (page 5). Moreover, we added a rationale of why it is important to investigate these separate dimensions (page 6): “Furthermore, investigating both the intrapersonal and interpersonal dimension of maladaptive perfectionism may clarify what aspect of maladaptive perfectionism is most relevant to program effectiveness, and should therefore be taken into account. This might especially be important in adolescents, as adolescence is a developmental phase characterized by the formation of identity, seeking independence from parents and increasing peer influence (Branje et al., 2021; Brown, 2004).” In the discussion section, we added a new paragraph to elaborate on the findings of the different dimensions (page 18): “Both the intrapersonal (i.e., self-criticism) and interpersonal (i.e., socially prescribed perfectionism) dimension of perfectionism were significant moderators, with similar effect sizes. This indicates that both dimensions are relevant to program effectiveness. Interestingly, parental pressure did not affect program effectiveness, whereas socially prescribed perfectionism, thus the belief that others – not only parents – hold perfectionistic expectations, did. Possibly, adolescents’ sensitivity to peer norms might be a crucial factor to explain these diverging findings. As peers and friends become more important during adolescence, peer influence and pressure could have a significant role in adolescents’ attitudes and behavior (Brown, 2004), and therefore also in program effectiveness. Future research in this area is warranted.”.
- Clear hypotheses are needed at the end of the Introduction session. The current writing is very vague and it is not clear what findings are anticipated from the study.
We added hypotheses at the end of the introduction: “It is expected that adolescents with higher problem severity benefit more from the program. No clear hypotheses are formulated for maladaptive perfectionism, as previous literature is inconsistent. Finally, it is expected that adolescents with higher parental pressure benefit less from the program.” (page 7).
- I appreciate the authors’ efforts in including gender as a potential covariate. I am wondering whether other covariates should be considered as well. For example, could students’ grade level be related to program efficacy as students may have different course/test loads across different school grades?
We included gender as a covariate because there were differences at T1 between the experimental and control group (see van Loon et al., 2023). Since there were no differences at T1 between the experimental and control group for educational level and school year (grade), we did not include this as a covariate in the analyses.
- There are a few typos in the manuscript, for example, a few places listed “program severity”.
In response to this comment and comment #4 of reviewer 1, we corrected these typos.
- I was wondering whether the data captures any of students’ peer contexts. Adolescence is the developmental stage when peers and friends become important socialization agents and peer influence and pressure could have outstanding role in adolescents’ emotions and behaviors. Given the current study does not find parental pressure as a moderator, peer context might be something to consider.
We agree with the reviewer that students’ peer context is important, especially in adolescence. Unfortunately, we do not have specific information on the adolescents’ peer context. Most items of the socially prescribed perfectionism subscale were broad, asking about others or other people (e.g., “There are people in my life who expect me to be perfect”), only two items asked about family or teachers specifically. Since parental pressure was not a significant moderator, it might be that other contexts are more important, such as peers. In response to this comment and comment #1, we added an extra paragraph to the discussion: “Both the intrapersonal (i.e., self-criticism) and interpersonal (i.e., socially prescribed perfectionism) dimension of perfectionism were significant moderators, with similar effect sizes. This indicates that both dimensions are relevant to program effectiveness. Interestingly, parental pressure did not affect program effectiveness, whereas socially prescribed perfectionism, thus the belief that others – not only parents – hold perfectionistic expectations, did. Possibly, adolescents’ sensitivity to peer norms might be a crucial factor to explain these diverging findings. As peers and friends become more important during adolescence, peer influence and pressure could have a significant role in adolescents’ attitudes and behavior (Brown, 2004), and therefore also in program effectiveness. Future research in this area is warranted.” (page 18).
- In Section 4.1, I would suggest revise the leading sentence to be more cautious. Although this study includes culturally diverse participants, I’m worried the sample size cannot be treated as large with ~120 participants.
We nuanced the first sentence of the strengths and limitations section. Although each group (experimental and control group) contained about 100 participants, the subsample of participants who attended more than half of the sessions contained only about 50 participants for the experimental group. We agree with the reviewer that we cannot treat the sample size as large, and removed this part of the sentence.

Round 2
Reviewer 2 Report
Comments and Suggestions for Authors
Thank you for the opportunity to review the revised manuscript “What Works for Whom? The Influence of Problem Severity, Maladaptive Perfectionism and Perceived Parental Pressure on the Effectiveness of a School-Based Performance Anxiety Program”. I appreciate authors’ efforts to address my comments and I think the quality of the manuscript has improved. I have a few remaining suggestions that I would like to share:
- In the abstract, I think the authors should present the sample characteristics of the final sample (i.e., 196) versus the larger number before excluding withdrawn participants.
- Across the Results section, I recommend keeping consistent number of digits after the decimal points, currently some results are presented with 3 decimal numbers and others with 2.
- In the Discussion section, I recommend integrating the last two paragraphs before section 4.1 as both paragraphs talk about why parental pressure was not related to program effectiveness. For example, the authors can talk about why parental pressure did not show anticipated effect (which is the later part of the first paragraph and the main body of the second paragraph) and then talk about explanations for these non-significant findings, and the potential effect of peer contexts that were not captured in this study. Similarly, the leading sentence of the first paragraph (i.e., Both the intrapersonal and interpersonal...) can be integrated with the previous paragraph as a concluding sentence.
Thank you for the hard work on this manuscript.
Author Response
Thank you for the opportunity to review the revised manuscript “What Works for Whom? The Influence of Problem Severity, Maladaptive Perfectionism and Perceived Parental Pressure on the Effectiveness of a School-Based Performance Anxiety Program”. I appreciate authors’ efforts to address my comments and I think the quality of the manuscript has improved. I have a few remaining suggestions that I would like to share:
- In the abstract, I think the authors should present the sample characteristics of the final sample (i.e., 196) versus the larger number before excluding withdrawn participants.
We agree with the reviewer, and now report the characteristics of the final sample in the abstract (page 2). We also changed the characteristics in the methods section, now representing the final sample as well (2.2. participants, page 8).
- Across the Results section, I recommend keeping consistent number of digits after the decimal points, currently some results are presented with 3 decimal numbers and others with 2.
We adjusted the number of decimal points to 2 for all statistics (except for percentages, we now report zero decimal places), in the text (Results section) and the (Supplementary) Tables (pages 11-14, pages 20-21).
- In the Discussion section, I recommend integrating the last two paragraphs before section 4.1 as both paragraphs talk about why parental pressure was not related to program effectiveness. For example, the authors can talk about why parental pressure did not show anticipated effect (which is the later part of the first paragraph and the main body of the second paragraph) and then talk about explanations for these non-significant findings, and the potential effect of peer contexts that were not captured in this study. Similarly, the leading sentence of the first paragraph (i.e., Both the intrapersonal and interpersonal...) can be integrated with the previous paragraph as a concluding sentence.
We agree with the reviewer that we can integrate the two paragraphs about parental pressure (before the strengths and limitations section) into one paragraph. We changed this accordingly (pages 18-19).